# Endothelial Cell Response to Combined Photon or Proton Irradiation with Doxorubicin

**DOI:** 10.3390/ijms241612833

**Published:** 2023-08-16

**Authors:** Teresa Bernardo, Anna Kuntze, Diana Klein, Feline Heinzelmann, Beate Timmermann, Cläre von Neubeck

**Affiliations:** 1Department of Particle Therapy, University Hospital Essen, University of Duisburg-Essen, 45147 Essen, Germany; teresa.bernardo@uk-essen.de (T.B.); beate.timmermann@uk-essen.de (B.T.); 2Gerhard Domagk Institute of Pathology, University Hospital Muenster, 48149 Muenster, Germany; anna.kuntze@ukmuenster.de; 3Institute of Cell Biology (Cancer Research), University Hospital Essen, University of Duisburg-Essen, 45147 Essen, Germany; diana.klein@uk-essen.de; 4West German Proton Therapy Centre Essen (WPE), 45147 Essen, Germany; feline.heinzelmann@uk-essen.de; 5West German Cancer Centre (WTZ), University Hospital Essen, University of Duisburg-Essen, 45147 Essen, Germany; 6Faculty of Physics, Technical University (TU) Dortmund University, 44227 Dortmund, Germany; 7German Cancer Consortium (DKTK), 45147 Essen, Germany

**Keywords:** endothelial cells, proliferation, migration, cell survival, proton beam radiotherapy, combined treatment, doxorubicin, additive effects

## Abstract

Surgery, radiotherapy, and chemotherapy are essential treatment modalities to target cancer cells, but they frequently cause damage to the normal tissue, potentially leading to side effects. As proton beam radiotherapy (PBT) can precisely spare normal tissue, this therapeutic option is of increasing importance regarding (neo-)adjuvant and definitive anti-cancer therapies. Akin to photon-based radiotherapy, PBT is often combined with systemic treatment, such as doxorubicin (Dox). This study compares the cellular response of human microvascular endothelial cells (HMEC-1) following irradiation with photons (X) or protons (H) alone and also in combination with different sequences of Dox. The cellular survival, cell cycle, apoptosis, proliferation, viability, morphology, and migration were all investigated. Dox monotreatment had minor effects on all endpoints. Both radiation qualities alone and in combination with longer Dox schedules significantly reduced clonogenic survival and proliferation, increased the apoptotic cell fraction, induced a longer G2/M cell cycle arrest, and altered the cell morphology towards endothelial-to-mesenchymal-transition (EndoMT) processes. Radiation quality effects were seen for metabolic viability, proliferation, and motility of HMEC-1 cells. Additive effects were found for longer Dox schedules. Overall, similar effects were found for H/H-Dox and X/X-Dox. Significant alterations between the radiation qualities indicate different but not worse endothelial cell damage by H/H-Dox.

## 1. Introduction

Blood vessels are composed of endothelial cells (EC) on the inner surface and perivascular cells on the outer surface [1]. Both cell types interact closely in normal tissue regulating perfusion, permeability, oxygenation, and blood flow. Tumor vasculatures develop either via angiogenesis (dominant mechanism) by the sprouting of ECs from existing blood vessels or via vasculogenesis by the formation of de novo blood vessels. To build new vessels, tumor cells stimulate the migration and the proliferation of ECs. The tumor vessel network is characterized by high disorganization, with immature and hyperpermeable vasculature resulting in low perfusion and increasing hypoxia [1]. Both oversupplied and undersupplied areas within tumors complicate the efficient administration of intravenous drugs in cancer therapy due to their disordered structure [2]. The effects of photon-based radiotherapy (XRT) on vasculatures are well described and reviewed in [3]. XRT has a dose-dependent impact on tumor vasculature, and it influences the response to anticancer therapies [1]. While low-dose radiation (<2 Gy) promotes angiogenesis, moderate and high-dose radiation have an inhibitory effect on angiogenesis. High-dose radiation (>10 Gy) additionally induces EC death and acute vascular damage, resulting in reduced blood perfusion and increased hypoxia [1,4]. Moderate doses of radiation (2–10 Gy), as used in this study, induce EC senescence or permanent cell cycle arrest. However, XRT-induced higher permeability is one strategy used to improve drug delivery [1,5]. Microvascular ECs are especially sensitive to ionizing radiation (IR). IR induces cell activation, and it increases barrier permeability and apoptosis mediated by p53 [3]. Overall, ECs and tumor microenvironment play an important role in cancer radiotherapy [4].

In contrast to XRT-induced changes in the vessel system, the effects of PBT are less understood. Relative to photons (X), protons (H) have an inverted depth dose profile (Bragg-peak curve) and deliver less dose to the tumor adjacent normal tissue [6]. By the energy modulating and layering of Bragg-peak curves, a uniform dose distribution is achieved in the tumor, which is the so-called Spread-Out Bragg-Peak (SOBP). For the tumor, a 10% higher relative biological effectiveness (RBE) of H in comparison to X is assumed in the clinic. The question of whether the 10% higher RBE translates to the vasculature is not known. However, it has been shown that H irradiation significantly inhibited vascular development in zebrafish, possibly due to increased vascular cell death via radical oxygen species (ROS) formation. ROS are the main regulators of IR-induced cell death through direct DNA damage [7]. Additionally, H irradiation inhibits the expression of pro-angiogenic factors and multiple angiogenesis-associated processes, including invasion and EC proliferation [8]. IR induce cell cycle arrest, senescence, and apoptosis in rapidly proliferating ECs within the radiation field [9]. PBT is increasingly used for patients with treatment-resistant tumors in order to deliver higher doses to the tumor, or for pediatric patients to spare the normal tissue.

Due to the favorable dose distribution, PBT becomes an important treatment option for soft tissue sarcoma (STS), i.e., malignant mesenchymal tumors that can occur all over the body and at all ages [10]. The first-line drugs of STS are doxorubicin (Dox), ifosfamide, and dacarbazine [11]. Due to the lack of clinical studies on the combined efficacy of chemotherapy and PBT, experimental investigations on both tumors and normal tissues are needed.

Dox is an anthracycline drug with severe cardiotoxic side effects, e.g., permanent damage of ECs. Dox leads to the inhibition of protein synthesis via p53 expression and VEGFR2 downregulation, which is associated with long-term senescence of ECs [12]. In Dox-induced cardiotoxicity (DIC), Dox stimulates the accumulation of ROS, which activates endothelial-to-mesenchymal (EndoMT) transition and dysregulated autophagy in ECs [13]. EndoMT is a biological process in which ECs lose their characteristic phenotype and transform into mesenchymal cells with a fibroblast-like appearance. Moreover, the endothelial damage in DIC leads to the disruption of the endothelial barrier function and increases vascular permeability, resulting in inflammation [14].

For tumors, particularly for STS, we have previously reported that Dox showed similar additive effects in combination with X or H irradiation dependent on treatment schedules [15]. In this study, we aim to characterize the effects of H and X irradiation alone or in combination with Dox treatment in a preclinical endothelial model (HMEC-1 cells) in a similar set up. The sequence of the combined treatments is modulated by applying Dox (i) only pre-, (ii) pre- and post-, or (iii) only post-irradiation to reveal the size of chemotherapy and radiation modalities in endothelial cell damage.

## 2. Results

### 2.1. Clonogenic Survival: Combined Dox Treatment Reduced Clonogenic Survival of HMEC-1 Cells More Efficently upon Prolonged Treatment

Colony formation assay (CFA) is one of the most important assays in radiation biology, and it determines cell survival following radiation [16]. CFA was performed following X or H irradiation (absorbed dose: 0–8 Gy) in order to define the radiosensitivity of HMEC-1 cells. Survival curves were fitted with the linear quadratic model (LQM) and showed no significant difference (Figure 1A). Based on the fit parameters, the RBE values were calculated (Figure 1B). Figure 1B shows that HMEC-1 cells exhibit in the low-dose region a higher sensitivity for H, and in higher dose regions exhibit a higher sensitivity for X irradiation. The chemosensitivity of HMEC-1 cells towards Dox (0–7.5 nM) were tested in three sequences (Figure 1C): Dox pre- (DoxA), pre- and post- (DoxB), and post-irradiation (DoxC). DoxA had a non-significant effect on cellular survival, while DoxB and DoxC significantly lowered the ability of colony formation (Figure 1D). There was no significant difference between DoxB and DoxC monotreatment. Similar effects were found for combined treatment with 4 Gy X or H irradiation (Figure 1E,F). Here, DoxB and DoxC treatment with 5 and 7.5 nM significantly reduced cell survival. We compared Dox monotreatment with combined Dox-IR treatment (Figure 1G,H). Only 5 nM DoxB in combination with 4 Gy X or H significantly lowered the ability to form colonies (Figure 1G).

### 2.2. Apoptosis: HMEC-1 Cells Are Chemoresistant for Dox Treatment Alone Independent of Treatment Schedule, but Sensitive for Combined Treatment with Radiation

Radiation [17] and Dox [18] can induce apoptosis or mitotic cell death. Flow cytometer analysis and PI staining determining apoptotic DNA-fragmentation was conducted for the first 96 h following X or H irradiation and 10 nM Dox treatment with different schedules. The subG1 population indicating apoptosis increased with time following radiation, but only X irradiation significantly increased the apoptosis level (Figure 2A). HMEC-1 cells were resistant to monotreatment with Dox (Figure 2B), and DoxA even showed significantly lower apoptosis level than control cultures.

A combined treatment of Dox-IR increased apoptosis over time (Figure 2C–F). Following X-Dox, treatment schedules DoxB or DoxC were the most effective (Figure 2C,D), while H-Dox increased apoptosis in all three schedules relative to 0 Gy controls (Figure 2E,F). To identify the potential influence of the radiation quality, matching X and H samples were compared, but no significant difference could be detected. The datasets were normalized to the respective dose (4 or 8 Gy), radiation quality (X or H), and time matching (48, 72, 96 h) samples to analyze potential additives or synergistic effects (Appendix A). With the exception of 4 Gy X-DoxC, no additive effects were found (Appendix A).

### 2.3. Cell Cycle Distribution: Accumulation of G2/M Population in HMEC-1 Cells Following Mono- or Combined Treatment with Radiation and Dox

Radiation and chemotherapy are DNA-damaging therapies and can induce transient or permanent cell cycle arrest, stopping the proliferation of damaged cells to provide an opportunity for DNA repair [19]. The effect of mono- or combined treatments on the cell cycle were analyzed after 48 h and 96 h (Figure 3). Relative to controls, HMEC-1 cells showed a dose-dependent and significant G2/M block following both radiation qualities, with small differences between X and H irradiation (Figure 3A). The G2/M cell fraction remained stable in the 4 Gy dose groups, but in the higher dose group, more cells accumulated in G2/M after 96 h (Figure 3A). Interestingly, the combined treatment with 4 Gy and any Dox treatment did not show a significant change in G2/M phase cell at 48 h, pointing towards an antagonistic effect as IR mono-treatment induced a G2/M block. At 96 h, both dose groups and radiation qualities increased the G2/M phase cell fraction. The maximum G2/M block was measured for 8 Gy X-DoxB (47.9%) at 96 h (Figure 3C). There is a trend in the data that following H-Dox, the G2/M block starts to resolve at 96 h post-treatment.

### 2.4. Proliferation: Prolonged Dox Treatment Combined with Radiation Have Additive Effects on Reduction of Proliferation Activity in HMEC-1 Cells

Giving the increased apoptosis rate and the prolonged G2/M phase arrest, the overall proliferation activity of HMEC-1 was analyzed (Figure 4). Following both radiation qualities, HMEC-1 cells showed a dose-dependent reduction in proliferation activity, which was most significant at 96 h post-treatment (Figure 4A). The treatment with Dox alone had no prolonged effect on proliferation at 96 h, but the most intense Dox treatment, DoxB, showed a significant decrease in proliferation during the observation period (Figure 4B). In combined treatment approaches, the proliferation activity was significantly reduced in a dose-dependent manner at 96 h in all conditions (Figure 4C–F). To identify the potential influence of radiation quality, time- and dose-matching X and H exposed samples were further compared (Appendix A). However, over the observation period (whole curve), no significant difference between X and H could be detected (Appendix A). To analyze potential additive or synergistic effects in combined treated samples, the data were normalized to the respective dose (4 or 8 Gy), radiation quality (X or H), and time-matching (48, 72, 96 h) samples (Appendix A). HMEC-1 cells could recover from the combined treatment with DoxA schedule, and no significant changes on proliferation activity were detected (Appendix A). In contrast, combined X/H-DoxB and X/H-DoxC showed a radiation dose-independent significant reduction at 96 h post-treatment, indicating additive effects. Interestingly, only samples exposed to 4 Gy showed additive effects over the whole observation period (Appendix A).

### 2.5. Cell Viability: X Radiation Alone and in Combination with Dox Treatment Increase Cell Viability at 72 h Post-Treatment

Post-mitotic, senescent, or proliferative inactive cells can still be viable and metabolically active. The viability of HMEC-1 cells was therefore measured via the WST-1 metabolic activity assay (Figure 5). The metabolic cell viability after IR monotreatment (Figure 5A) or Dox monotreatment (Figure 5B) was not significantly altered relative to the control cells, with the exception of 8 Gy X at 96 h post-treatment. Interestingly, all Dox schedules in combination with 4 Gy X were not significantly changed (Figure 5C), but 8 Gy X-DoxA-C increased the metabolic cell viability significantly (Figure 5D). Following H irradiation, there was no change in metabolic cell viability, with the exception of 4 Gy DoxA (Figure 5E,F). The viability curves for radiation as mono- or combined treatment have a district shape with a very high metabolic activity at 72 h post-treatment, which could not be detected following Dox monotreatment. This effect is more pronounced in X exposure samples and to higher radiation doses as well as to prolonged Dox schedules (Appendix A). The datasets were normalized to the respective dose (4 or 8 Gy), radiation quality (X or H), and time-matching (48, 72, 96 h) samples to reveal potential additive effects. However, no additive effects on the enhanced metabolic viability of the combined treatment over the monotreatment could be detected.

### 2.6. Cell Motility: X Radiation Reduced Motility of HMEC-1 Cells Stronger Than H Radiation

One of the hallmarks of early angiogenesis is EC migration [20]. Here, the lateral migration was determined via “wound closure” of an induced scratch following mono- or combined treatments (Figure 6, Appendix A). Relative to untreated controls, no effect on migration was detected for all of the treatment schedules, with the exception of 4 Gy H-DoxB (Figure 6). The maximum migration speed was calculated from the exponential phase of the motility curve between 3 and 24 h post-scratch induction (Figure 6G). Applied treatments reduced the migration speed, and the minimum speed was detected following 8 Gy H-DoxC (Figure 6G). In order to identify the potential influence of the radiation quality, time and dose matching X- and H-irradiated samples were compared (Appendix A). A significant reduction in migration was found for 8 Gy X alone and for 8 Gy X-DoxB relative to the matching H samples (Appendix A). The datasets were normalized to the respective dose (4 or 8 Gy), radiation quality (X or H), and time-matching samples (Appendix A) to identify potential additive effects. These were detected for 4 Gy X-DoxC, 4 Gy H-DoxB, and 8 Gy H-DoxC in reduced cell motility (Appendix A).

### 2.7. Cell Morphology Analysis Post-Treatment: Combined Treatment with Radiation and Prolonged Dox Treatment Show Additive Effects on Morphology

Based on the images for the migration assay, the morphological changes upon treatment were studied (Appendix A). Under the given cell culture conditions, untreated HMEC-1 cells showed a small and polygonal shape. Cells clustered and exhibited a cobblestone-like appearance. The nuclei were small and isomorphic. In between, very few cells had short cytoplasmic processes and showed a spindled morphology. Upon X irradiation, a significant fraction of HMEC-1 cells died, causing the surviving population to lose their structural organization. Approximately 40% of surviving cells changed from a polygonal shape to a spindled morphology. The cytoplasm of most HMEC-1 cells seemed to be enlarged, and cells possessed expanded nuclei. Following H radiation, HMEC-1 cells showed similar morphological alterations compared to X irradiation, with most of the spindled-shaped cells growing in a whirling architecture. DoxA monotreatment induced no extensive morphological differences compared to the untreated controls, while a few more spindled-shaped cells became present. DoxB and DoxC monotreatment showed time-dependent changes. HMEC-1 cells developed anisomorphism due to enlargement and cell shape variations. The grade of anisonucleosis increased. IR-DoxB and IR-DoxC led to similar morphological changes relative to IR monotreatment but to a greater extent. However, following IR-DoxB/C treatment, HMEC-1 cells lost their structural organization due to massive cell death. The enlargement and anisonucleosis exceeded the alterations in the Dox monotreatment group, indicating additive effects. Taken together, the observed morphological changes following therapy led to a fibroblast-like appearance of endothelial cells and, therefore, strongly suggest the induction of EndoMT.

## 3. Discussion

PBT is increasingly used for pediatric cancer patients, recurrent or treatment resistant tumors, and tumors adjacent to critical normal tissue structures, which are all situations where the normal tissue needs to be spared maximally by radiotherapy. Up until today, combined treatment strategies are the standard of care for many entities. However, established chemo-radiation therapies are generally translated from XRT to PBT without further clinical testing, and preclinical studies need to fill this gap in knowledge. A frequently PBT-treated tumor entity is STS, which mainly occurs in pediatric patients [10]. The first-line chemotherapy in this vulnerable patient group is the anthracycline Dox, which acts via apoptosis but which can also induce DIC mediated by EC damage.

Therefore, in this study, the effect of both X and H irradiation as mono- or combined treatment with Dox in three clinically relevant schedules were analyzed on cell survival, apoptosis, G2/M cell cycle arrest, metabolic viability, proliferation, migration, and morphological changes in human HMEC-1 endothelial cells. Overall, no significant difference in clonogenic cell survival following X or H was detected. DoxA had no effect, but DoxB and DoxC significantly lowered the survival after both radiation qualities (Figure 1). The apoptotic cell fraction was dependent on dose, time, and radiation quality, with higher apoptotic rates following X irradiation or X-Dox schedules (Figure 2). HMEC-1 cells induced a G2/M cell cycle arrest, which was more pronounced for later time points and higher doses. Interestingly, the combined treatment with 4 Gy-Dox did not show a significant change in G2/M cell cycle arrest at 48 h, which pointed towards an antagonistic effect as IR monotreatment induced a G2/M block (Figure 3). IR or IR-Dox treatment reduced the proliferation activity, but no prolonged effect of Dox monotreatment was seen (Figure 4). The metabolic viability of HMEC-1 cells was not altered following Dox monotreatment, but radiation quality effects were seen with higher metabolic viability following X-Dox relative to H-Dox (Figure 5). IR, Dox, or IR-Dox (Figure 6) did not affect cellular motility but morphological changes after IR or IR-DoxB/C point towards EndoMT processes (Appendix A). Few additive effects were found, and when they were, mainly for the longer Dox treatment schedules DoxB and DoxC (Appendix A). Differences in radiation quality were observed rarely but were found for proliferation (Appendix A), metabolic viability (Appendix A), and motility (Appendix A).

In a clinical setting, Dox is generally applied in an adjuvant or neoadjuvant manner as an intravenous infusion. The variation of blood clearance of Dox is high, but it can be assumed that Dox is still present in the tissue at the time of radiotherapy [21]. Here, DoxA (pre-treatment) is less damaging relative to the prolonged DoxB/C treatment schedules ((pre-)/post-treatment), particularly when combined with IR. However, there are no major differences in H-IR relative to X-IR. Clinically, Dox is given in multiple cycles, and the cytotoxic effect will ultimately increase. As one of the limitations of this study, it should be noted, that Dox was only applied once, and HMEC-1 cells were exposed to a single fraction of radiation. This is due to the fact that 2D cell cultures do not have the capacity for multiple repeated treatments. To capture the effect of Dox cycles or IR fractionation, in vivo or in ovo experiments are needed. A further limitation is the EC monoculture. The tumor microenvironment influences the vascular structure and organization within but also adjacent to the tumor. The portion of normal non-influenced EC that will be exposed to radiation is low, but the systemic Dox treatment will affect all ECs. It was shown that even transient exposure of ECs to Dox leads to long-lasting VEGFR2 downregulation, p53 expression, and inhibited global protein synthesis [12]. To capture the crosstalk between the tumor and EC, more complex 3D cultures in in vivo or in ovo models are needed.

The cellular response of both cell types, tumor cells and endothelial cells, is important for the success of anticancer therapies. It is obvious that eradicating tumor cells contributes to the local control rate, but EC killing did not further improve control rates in primary sarcoma treated by stereotactic body radiation therapy [22]. It is known that H irradiation suppresses angiogenic signalling in both cancer and normal cells, inhibits cell invasion, and modulates cellular proliferation as well as survival [8]. Tumor vessels can be normalized to improve drug and oxygen distribution, and, finally, will strengthen the radiation efficacy [2]. Radiosensitizers are being developed that target the tumor endothelium. For example, microtubule stabilising agents are tested to sensitize tumor cells via cell cycle arrest in the most radiosensitive G2/M phase or reoxygenation by causing damage directly or indirectly EC [23].

It is known that proliferating HMEC-1 cells undergo two phases of death after high-dose radiation with 15 Gy: first, an early pre-mitotic apoptosis, and, second, a delayed DNA damage-induced mitotic death [17]. We could show that HMEC-1 cells are radiosensitive to X and H irradiation, but no significant differences between X and H irradiation were seen (Figure 1A). The RBE_α_ of 1.2 is slightly higher than the clinically assumed 10% higher effectiveness of H [6]. This indicates a slightly higher sensitivity for H relative to X in the low-dose region, which may be important in the context of fractionated dose application [6]. The H experiments were performed in the middle of the SOBP. Additional exposure scenarios, which should be tested, are in the distal fall-off of the Bragg Peak, where a higher effectiveness of H is assumed, and in the entrance plateau region, where a lower effectiveness of H is expected. Both treatment situations are clinically relevant and include normal EC. Causal for the expected differences is the changing linear energy transfer (LET) along the H beam track, which translates to a variable RBE [24]. If the variable LET is influencing the effect of combined treatment with IR-Dox, this needs to be investigated.

Dox and ifosfamide remain the gold standard in STS treatment [11], but Dox is also approved for leukemia and for breast, gastric, lung, thyroid, and bladder cancer as well as for different types of lymphoma [25]. Dox monotreatment was generally less effective than IR monotreatment in causing a biological effect. However, additive effects were found in pre-clinical STS for Dox-IR, making it an effective clinical treatment [15]. However, the cytotoxic effects of Dox impact the tumor endothelium and EC [26]. Apoptosis is the main mechanism of action of Dox, and is one of the death mechanisms following IR. The TP53 status of the tumor is therefore crucial for clinical outcomes, while ECs are generally the TP53 wild-type [18]. Here, X monotreatment induced significant amounts of apoptosis while H monotreatment had no significant effect (Figure 2A). However, in combined treatments, higher apoptotic rations were found for longer Dox schedules and H irradiation (Figure 2E,F), suggesting a higher potential for H-Dox in damaging the endothelium.

Radiation and Dox treatment are known to influence the cell cycle distribution, leading to a cell cycle arrest in the G2/M phase [23]. In this study, we found significant accumulation in G2/M following IR monotreatment and 8 Gy-IR-Dox but not 4 Gy-IR-Dox at 48 h post-treatment, which suggests an antagonistic effect. At a later time point, all treatment schedules led to a significant G2/M arrest (Figure 3). This observation is interesting as the G2/M phase is the most sensitive phase of the cell cycle for IR damage. Moreover, DNA damage repair takes place in this cell cycle arrest [19]. Current clinical practice applies 2 Gy per radiation fraction, but the current treatment trends are towards hypofractionated regimes, meaning more dose-per-fraction and overall shorter treatment times [27]. Our findings suggest that the dose-per-fraction in combined treatment schedules could critically influence the damaging potential to EC. As mentioned above, the accumulating cytotoxic effect in fractionated treatments need to be investigated in more complex models.

With proliferation rates approaching zero under steady-state conditions, ECs are among the most quiescent cells in the body. Only after stimulation via injury, inflammation, or malignant growth, EC initiate mitosis in order to regain tissue integrity [20]. Under experimental conditions, EC are actively proliferating. Upon IR-monotreatment, HMEC-1 cells showed a dose-dependent reduction in proliferation, while Dox monotreatment had only minor effects. The combination of IR-Dox reduced proliferation significantly and in an additive manner for DoxB/C (Figure 4C–F, Appendix A) as well as in a radiation quality-dependent manner (Appendix A). These additive effects should be understood as in vitro synergy, which differs from therapeutic synergy [28]. It is known that H irradiation alone supresses angiogenic signalling in cancer and normal cells via inhibition of cell invasion, and modulation of cellular proliferation and survival in vitro [8]. In vivo experiments in zebrafish embryos could also demonstrate that H irradiation significantly inhibits vascular developments, possibly due to increased vascular cell death via reactive oxygen species (ROS) formation [7]. Our data strongly supports the need for further combined treatment studies with H irradiation to minimize vascular damage due to unfavourable treatment combinations.

In contrast to the proliferation assay, the metabolic viability assay showed no reduction of HMEC-1 cell viability (Figure 5). Strikingly, HMEC-1 cells increased the metabolic activity at 72 h post-X or post-X-Dox treatment by a factor of two, while H and H-Dox remained at the control level. It was shown that a moderate dose of X (2–10 Gy) induced EC senescence or permanent cell-cycle arrest via ROS production. Senescent cells can produce more ROS, which induces DNA damage, potentially leading to EC death [1]. Furthermore, senescent cells display a high metabolic activity, acquiring a more glycolytic state. This metabolic shift leads to increased AMP/ATP and AD/ATP ratios and subsequently to AMPK activation, and then, finally, to p53-mediated growth arrest [29]. Our findings are in line with the onset of senescence in HMEC-1 cells following X exposure, showing a cessation of proliferation, initiating cell cycle arrest in G2/M, and increasing metabolic viability. H-exposed HMEC-1 also reduces the proliferation activity and induces cell cycle arrest, but no increase in metabolic activity was seen. Moreover, there is a trend in the data that HMEC-1 cells resolve the G2/M block at 96 h post-IR (Figure 3). This could be indicative that X/X-Dox induce more senescence than H/H-Dox, which warrants further analysis.

Overall HMEC-1 cells have a low motility (Figure 6, Appendix A), which was only significantly lowered following 8 Gy X (Figure 6A); however, additive effects could be identified for different combinations of prolonged Dox treatment and IR. In addition to apoptosis, Dox damages the cellular membrane, causing a reduction of motility [18]. Morphological changes in HMEC-1 cells were seen for both radiation qualities and also in combination with Dox. Intensification of treatment led to more pronounced morphological effects, which showed classic features of EndoMT (Appendix A), where ECs lose their characteristics of cell differentiation and transform into mesenchymal cells with a fibroblast-like phenotype. In this process, ECs lose their adhesive properties and gain migratory and invasive characteristics with increased vascular permeability. EndoMT is a part of the molecular mechanisms of endothelial remodelling under Dox treatment [14]. In preliminary protein analysis, we could observe increased expression of mesenchymal markers in HMEC-1 upon combined treatment. However, further investigations should be performed, including in in vivo or in ovo tumors.

In summary, combined treatment with Dox reduced clonogenic survival and proliferation of HMEC-1 cells more efficiently upon prolonged treatment and showed additive effects on cell morphology. HMEC-1 cells are chemoresistant for Dox monotreatment alone, independent of treatment schedule, but they are sensitive to combined treatment regarding the induction of apoptosis and G2/M cell cycle arrest. In particular, the cell cycle arrest following combined treatments should be further investigated, providing clinical development to hypofractionation. The increase in cell metabolic viability at 72 h post-X/X-Dox is remarkable, which, in the overall view of the data, is indicative for the onset of senescence. X irradiation reduced the motility of HMEC-1 cells stronger than H irradiation, and morphological changes point towards EndoMT. This study is limited by EC monocultures and single fraction/Dox cycle, though, and to translate these findings in a more clinically relevant setting, additional experiments with more complex 3D cultures and tumors will be needed.

In conclusion, X/X-Dox as well as H/H-Dox induce EC damage, but partially different cellular processes are activated, and this needs further investigation. Alternative combined treatment strategies with H irradiation should also be considered as PBT spares normal tissue more effectively than XRT. A promising strategy in anticancer therapy is a switch-on prodrug. Such drugs can be locally activated via IR, and they hold the potential of reducing treatment associated side effects while being equally or even more potent in controlling the tumor locally. A modified Dox pro-drug has now been developed [30], and it should be tested in combination with PBT.

## 4. Materials and Methods

### 4.1. Cell Culture

The human dermal microvascular endothelial cells (HMEC-1), originally established by Ades and colleagues [31], were obtained from ATCC (CRL-3243). Cells were grown in M199 medium (Thermofisher scientific, Waltham, MA, USA) supplemented with 10% (*v*/*v*) fetal bovine serum, penicillin-streptomycin (100 U/mL), and 15 mg endothelium cell growth supplement (Sigma-Aldrich, St. Louis, MI, USA). Cells were maintained at 37 °C and 5% CO_2_ in a humidified incubator.

### 4.2. Photon Irradiation

Photon irradiation, hereafter referred to as X, was performed using the Isovolt 320-X-ray machine (Seifert–Pantak, East Haven, CT, USA) at 320 kV, 10 mA, with a 1.65-mm aluminum filter. Details of this are described elsewhere [15].

### 4.3. Proton Irradiation

Proton irradiation, hereafter referred to as H, was performed with an IBA ProteusPlus proton therapy system (IBA PT, Louvain-la-Neuve, Belgium) at the West German Proton Therapy Centre Essen (WPE). Spread-Out Bragg Peak (SOBP) irradiation was performed as published previously [15]. In brief, SOBP consisted of five energy layers (118.8–129.9 MeV). HMEC-1 cells in multi-well plates were irradiated in the middle of the SOBP with absorbed physical doses of 1, 2, 4, 6, or 8 Gy. The sample surface was in the isocenter on the treatment couch, with a gantry angle of 0°.

### 4.4. Doxorubicin Treatment

The anthracycline drug doxorubicin (Dox) (2 mg/mL, Medac GmbH, Wedel, Germany) was obtained from and prepared by the pharmacy of the University Hospital Essen. HMEC-1 cells were treated with Dox and diluted in PBS or cell culture medium in different sequences––DoxA: 3 h pre-irradiation; DoxB: 3 h pre-irradiation and refreshed within 1 h post-irradiation until the end of the experiment; or DoxC: within 1 h post-irradiation until the end of the experiment (DoxC) (Figure 1C).

### 4.5. Colony Formation Assay (CFA)

For the clonogenic survival, HMEC-1 cell were pre-seeded 7 h prior to radiation in triplicates in 6-well plates. Cells were treated with Dox-containing culture medium. Following irradiation, the media of all samples were changed with medium or Dox-containing medium. The colonies were fixed after eight days, stained using 0.3% crystal violet dye (Roth, Karlsruhe, Germany) in 70% Ethanol for 10 min at RT, rinsed with water, and air dried. Colonies with 50 cells were scored as surviving. Exemplary images of CFA are shown in Appendix A.

### 4.6. Flow Cytometry Analysis

Twenty-four hours before treatment, HMEC-1 cells were plated in 6-well plates. Propidium iodide (PI) staining and flow cytometry analysis for apoptotic DNA-fragmentation (subG1 population) was performed 48, 72, or 96 h post-treatment. HMEC-1 cells were incubated for 15–30 min at RT with a staining solution (0.1 M Tris, 0.1 M NaCl, 5 mM MgCl2, 0.05% Triton X-100 (all Roth, Karlsruhe, Germany)), 62 µg/mL of RNaseA (AppliChem, Darmstadt, Germany), and 40 µg/mL PI (Sigma-Aldrich, St. Louis, MI, USA) [32]. Subsequently, samples were analyzed by flow cytometry (FACS Calibur, Becton Dickinson, Heidelberg, Germany; FL-2) [33]. Cell cycle phase distribution was analyzed with Kaluza software Version 2.1 to identify the subG1 population (apoptotic DNA-fragmentation, whole population), and, in a second step, the living cell population (G1, S, G2/M phase) was investigated for a G2/M arrest [15]. Statistical analysis was performed in GraphPadPrism Version 8.3.0. Exemplary histograms of flow cytometry are shown in Appendix A.

### 4.7. Cell Viability and Proliferation Assay

To quantify the metabolic viability and proliferation activity of HMEC-1 cells, the WST-1 reagent (in PBS 1:3, Roche, Rotkreuz, Schweiz) in combination with crystal violet (CV) stain were used as described elsewhere [15]. The metabolic viability (WST) and proliferation (CV) data was normalized to 0 Gy or 0 nM controls.

### 4.8. Migration Assay

The migratory potential of cells was investigated 48 h post-treatment with a migration assay as described elsewhere [15]. In brief, images were taken 0, 3, 6, 9, 24, and 48 h post-scratch induction and analyzed with ImageJ plugin Wound_healing_size_tool_updated (Wayne Rasband, National Institutes of Health, USA). Due to treatment associated cell loss following 8 Gy radiation, automated image analysis could not be applied, and images were analyzed manually. Simple linear regression between two time points (3–24 h) were calculated and the maximum motility speed was estimated from the slope in the steep part of the curve. Additionally, a pathologist evaluated the morphological changes based on the images of the migration assay. Exemplary images of the migration assay are shown in Appendix A.

### 4.9. Data Analysis and Statistics

Data analysis and statistics were performed according to Bernardo et al. (2023) [15]. Cell survival and dose response data were fitted with the linear quadratic equation:$$SF=e^{-(\alpha D+\beta D^{2})}$$
where *SF* denotes the surviving fraction of cells at dose *D* with curve fitting parameters *α* and *β*. Nonlinear regression analysis was performed on survival curves using GraphPad Prism, version 8.3.0. *RBE* values for protons were calculated relative to 320 kV X-rays according to:$$RBE\;SF=\frac{D^{X\;SF}}{D^{H\;SF}}$$
where *RBE SF* is the *RBE* at a selected survival level (*SF*), and where *DX SF* and *DH SF* are the *X* and *H* dose for an iso-effect, respectively.

All data points represent at least three replicates with error bars representing standard deviation (SD), and statistical analyses were performed with GraphPadPrism 8.3.0. All of the presented data were normalized to the experiment, time, and treatment matching controls. SD for the controls of each assay were calculated. For the CFA, the plating efficiency (PE) was calculated. The corresponding SD represents the relative mean of the PEs. For subG1 levels (apoptosis) and cell cycle phase, SD was calculated from the mean of relative subG1 or cell cycle phase levels. For cell viability and proliferation assay, measurements were normalized to 0 Gy control and the corresponding SD was calculated from the relative mean of the measurements. For the migration assay, SD was calculated from the mean of relative motility. The significant level was determined by unpaired (curve comparison) or paired *t*-test (data point comparison) with *p* values > 0.05 (not significant, ns), <0.05 (*), <0.01 (**), and <0.001 (***) were considered statistically significant.

## Figures and Tables

**Figure 1 ijms-24-12833-f001:**
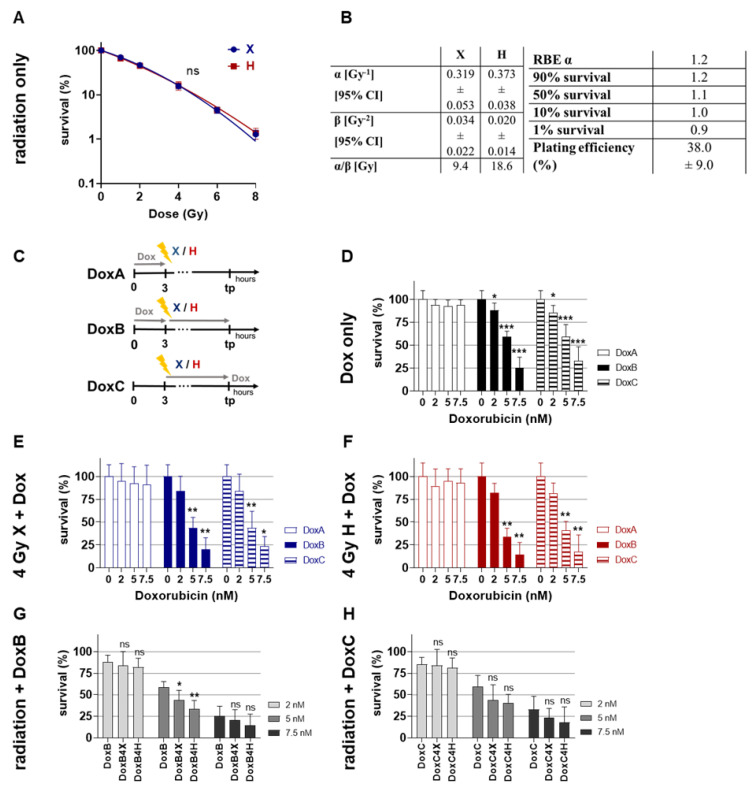
Colony formation assay. Clonogenic survival of HMEC-1 cells following (**A**) X radiation (blue) or H radiation (red) alone, fitted with the linear-quadratic model. (**B**) Table summarizing the fit parameter of the survival curves shown in (**A**), the maximum RBEα, the RBE values to survival levels of 90, 50, 10, and 1%, and the plating efficiency of the cell models. (**C**) Summary of doxorubicin (Dox) treatment schedules. DoxA: 3 h pre-(mock) irradiation followed by media exchange without Dox. DoxB: 3 h pre-(mock) irradiation followed by media exchange containing Dox and Dox exposure until the end of the experiment. DoxC: (mock) irradiation followed by media exchange containing Dox and Dox exposure until the end of the experiment. Mock Dox treatments (medium without Dox) were performed for all conditions. (**D**) Dox treatment alone or in combination with (**E**) 4 Gy X irradiation or (**F**) 4 Gy H irradiation. (**G**) DoxB treatment alone or in combination with 4 Gy X or H. (**H**) DoxC treatment alone or in combination with 4 Gy X or H. Dox was applied according to (**C**). Samples were normalized to matching 0 nM (+radiation) controls. n ≥ 3, statistical analysis: (**A**) paired *t*-test for the whole curve comparing X vs. H. (**D**–**F**) Unpaired *t*-test comparing mono-/combined treatment vs. matching 0 nM control. (**G**,**H**) Unpaired *t*-test comparing monotreatment with Dox vs. combined treatment. *p* values: > 0.05 (not significant, ns), <0.05 (*), <0.01 (**), and <0.001 (***).

**Figure 2 ijms-24-12833-f002:**
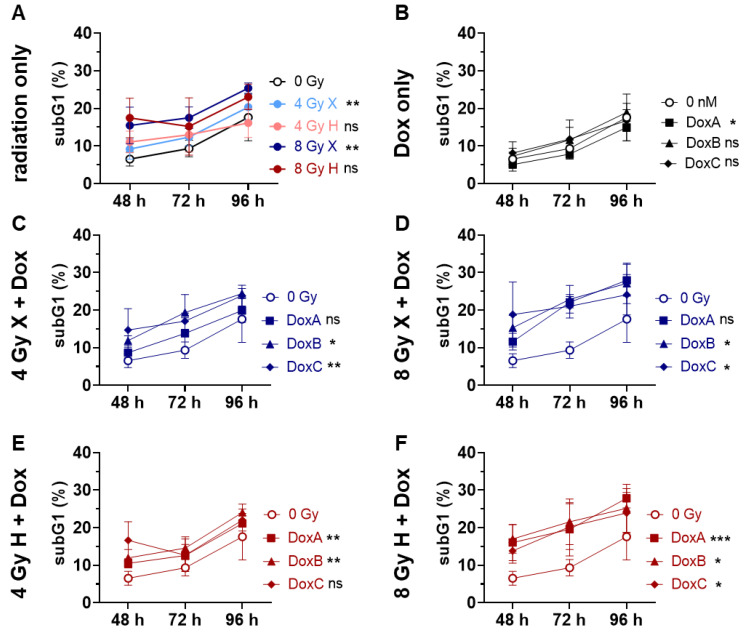
Flow cytometry comparing relative subG1 phase proportion (subG1) of the whole cell population (subG1, G1, S and G2/M phase) of 0 Gy control and treatment of HMEC-1 cells following (**A**) radiation only with 4 and 8 Gy X (light/dark blue) or 4 and 8 Gy H (light/dark red); (**B**) Dox treatment with 10 nM DoxA (pre), DoxB (pre and post), or DoxC (post); Dox treatment schedule details in Figure 1C. (**C**–**F**) combined treatment with DoxA, DoxB, or DoxC; and (**C**) 4 Gy X (blue), (**D**) 8 Gy X (blue), (**E**) 4 Gy H (red), or (**F**) 8 Gy H (red). n ≥ 3, Statistical analysis: paired *t*-test for whole curve comparing treatment vs. 0 Gy control. *p* values: > 0.05 (not significant, ns), <0.05 (*), <0.01 (**), and <0.001 (***).

**Figure 3 ijms-24-12833-f003:**
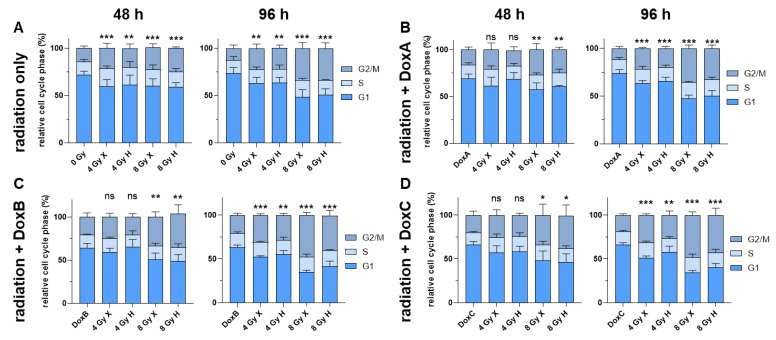
Flow cytometry comparing the relative cell cycle phase (G1 + S + G2/M = 100%) of HMEC-1 cells at 48 and 96 h following (**A**) 4 and 8 Gy X or H radiation only. (**B**–**D**) combined treatment with radiation and (**B**) DoxA (pre), (**C**) DoxB (pre and post), and (**D**) DoxC (post). Dox treatment schedule details in Figure 1C. n ≥ 3, Statistical analysis: unpaired *t*-test for each timepoint comparing treatment vs. matching control of G2 phase. *p* values: > 0.05 (not significant, ns), <0.05 (*), <0.01 (**), and <0.001 (***).

**Figure 4 ijms-24-12833-f004:**
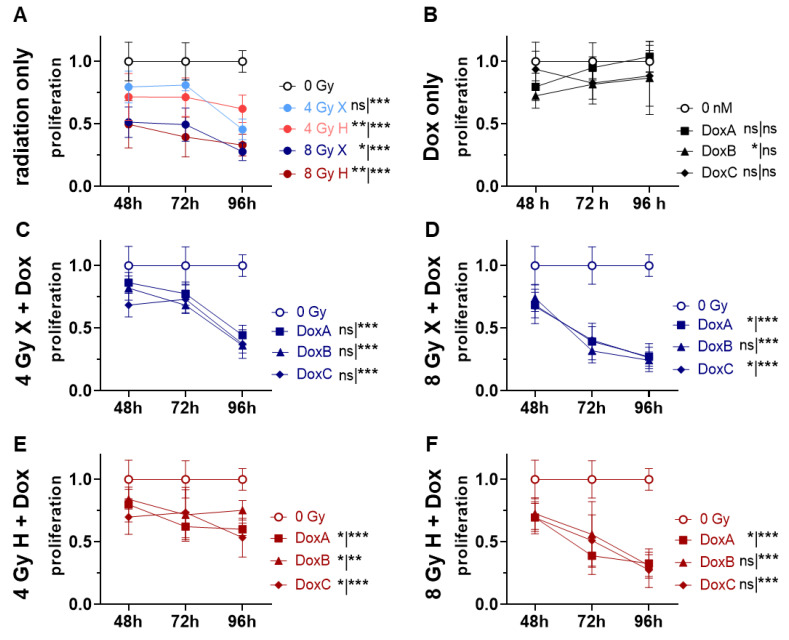
Proliferation assay comparing relative number of proliferating cells (proliferation; normalized to matched control) of 0 Gy control vs. treatment of HMEC-1 cells following (**A**) radiation only with 4 and 8 Gy X (light/dark blue) or 4 and 8 Gy H radiation (light/dark red). (**B**) Dox treatment with 10 nM DoxA (pre), DoxB (pre and post), or DoxC (post). Dox treatment schedule details in Figure 1C. (**C**–**F**) combined treatment with DoxA, DoxB, or DoxC, and (**C**) 4 Gy X (blue), (**D**) 8 Gy X (blue), (**E**) 4 Gy H (red), and (**F**) 8 Gy H (red). n ≥ 3, Statistical analysis: paired *t*-test for whole curve or unpaired *t*-test for 96 h timepoint shown as (whole curve|96 h) comparing treatment vs. 0 Gy control. *p* values: > 0.05 (not significant, ns), <0.05 (*), <0.01 (**), and <0.001 (***).

**Figure 5 ijms-24-12833-f005:**
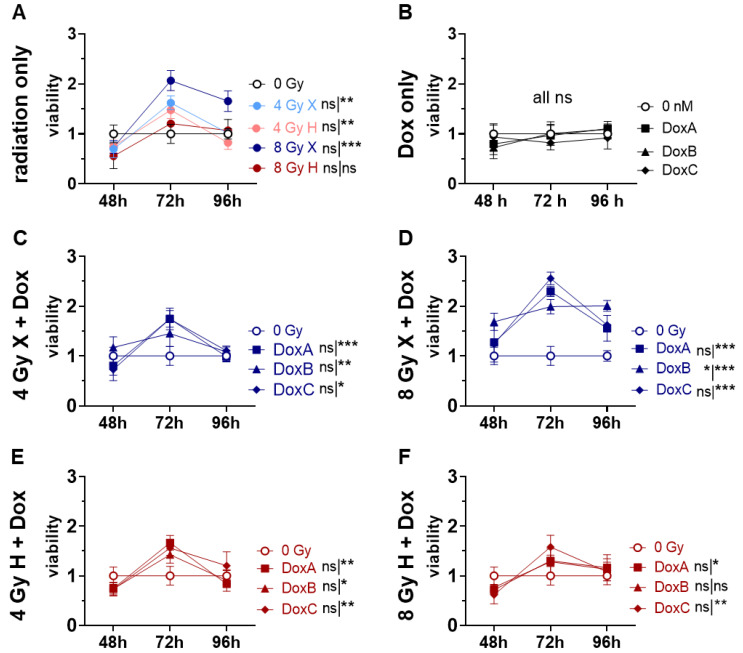
Cell viability assay comparing relative number of viable cells (viability; normalized to matched controls) of 0 Gy control vs. treatment of HMEC-1 cells following (**A**) radiation only with 4 and 8 Gy X (light/dark blue) or 4 and 8 Gy H radiation (light/dark red). (**B**) Dox treatment only with 10 nM DoxA (pre), DoxB (pre and post), or DoxC (post). Dox treatment schedule details in Figure 1C. (**C**–**F**) combined treatment with DoxA, DoxB, or DoxC, and (**C)** 4 Gy X (blue), (**D**) 8 Gy X (blue), (**E**) 4 Gy H (red), and (**F**) 8 Gy H (red). n ≥ 3, Statistical analysis: paired *t*-test for whole curve or unpaired *t*-test for 72 h timepoint shown as (whole curve|72 h) comparing treatment vs. 0 Gy control. *p* values: > 0.05 (not significant, ns), <0.05 (*), <0.01 (**), and <0.001 (***).

**Figure 6 ijms-24-12833-f006:**
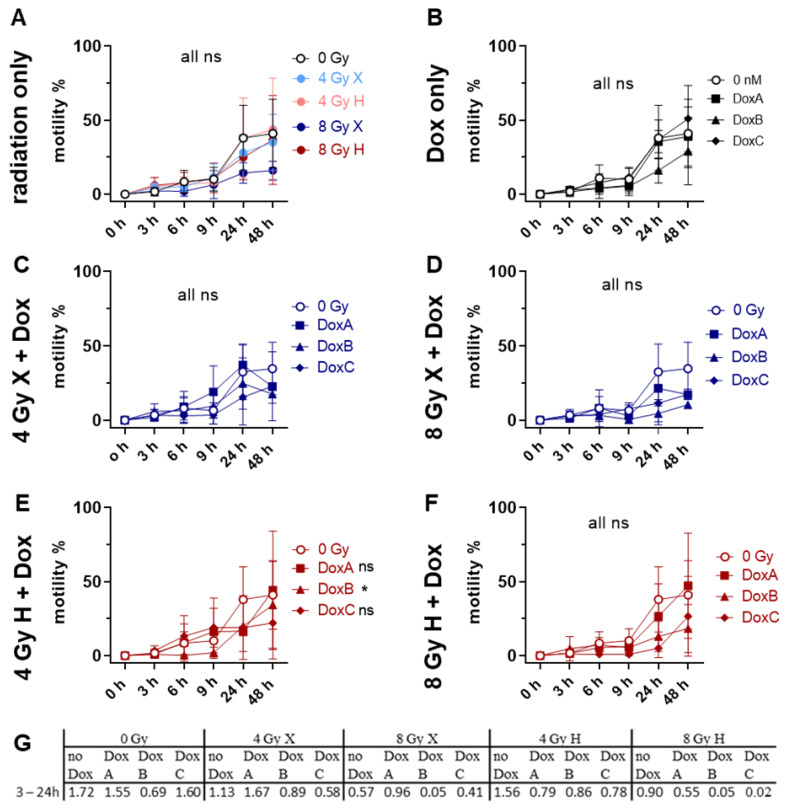
Migration assay comparing motility of 0 Gy control vs. treatment of HMEC-1 cells following (**A**) radiation only with 4 and 8 Gy X (light/dark blue) or 4 and 8 Gy H (light/dark red) radiation. (**B**) Dox treatment with 10 nM DoxA (pre), DoxB (pre and post), or DoxC (post). Dox treatment schedule details in Figure 1C. (**C**–**F**) Combined treatment with DoxA, DoxB, or DoxC, and (**C**) 4 Gy X (blue), (**D**) 8 Gy X (blue), (**E**) 4 Gy H (red), and (**F**) 8 Gy H (red). (**G**) Maximum migration speed extracted from the exponential phase of the curve via linear regression. n ≥ 3, Statistical analysis: paired *t*-test for whole curve comparison: treatment vs. 0 Gy control over the whole observation period of 0–48 h. *p* values: >0.05 (not significant, ns) and <0.05 (*).

## Data Availability

The original contributions presented in the study are included in the article/Appendix A. Further inquiries can be directed to the corresponding author.

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
