# Peer review of "Endothelial Cell Response to Combined Photon or Proton Irradiation with Doxorubicin"

_ijms, 2023, doi:10.3390/ijms241612833_

Round 1
Reviewer 1 Report
Dear authors,
An interesting and relevant article is devoted to the study compares the cellular response of human microvascular endothelial cells (HMEC-1) following irradiation with photons or protons alone, and in combination with different sequences of doxorubicin. It was shown that an doxorubicin monotreatment had minor effects on all endpoints. Both radiation qualities alone and in combination with longer doxorubicin schedules significantly reduced clonogenic survival and proliferation, increased the apoptotic cell fraction, induced a longer G2/M cell cycle arrest, and altered the cell morphology towards endothelial-to-mesenchymal-transition (EndoMT) processes. Radiation quality effects were seen for metabolic viability, proliferation, and motility of HMEC-1 cells. Additive effects were found for the longer doxorubicin schedules. Overall, similar effects were found for protons / protons doxorubicin and photons / photons - doxorubicin. Significant alterations between the radiation qualities indicate different but not worse endothelial cell damage by protons / protons - doxorubicin.
The data presented in the studies of the literature review are up-to-date and meet the purpose.
The presented studies are modern and meet the purpose.
The article presents data with arguments logically discussed.
References correspond to the topic and is logically justified.
The figures are informative and correspond to the description in the article.
Best regards,
Reviewer

Author Response
Dear reviewer,
Thank you very much for this positive feedback. We are very pleased that our research results were of interest to you. Thank you for taking the time to read our manuscript.
Best regards,
Teresa Bernardo & Cläre v. Neubeck for the authors
Reviewer 2 Report
The manuscript was aimed to examine the potential effects of the photon or proton radiation combined with doxorubicin treatment on the endothelial cells (ECs) proliferation, survival, and cell cycle distribution. THe authors also examined the morphological changes to highlight the potential impact of the treatment on the endothelial-to-mesenchymal-transition.
In gerenral, the manuscript is well prepared and illustrates the additive (as was expected) effects of the combined therapies, whereas a chemotherapeutic agent doxorubixin used alone has minor effects for the all EC characteristics indicated above.
I have the following recomendations regarding this manuscript:
1) I suggest the authors to show the original FACs data to ilustrate an impact of the treatments on apoptosis, and cell cycle distribution. It can be also as an example of one of the treatment schedules shown in the Supplemenrtary file. Similary, the images with the plates illustrating the changes in the numbers of colonies can be included into the Supplementary Figure.
2) The expression of EMT markers (E/N-cadherin, vimentin, etc.) is highy desirable to reveal whether the treatment schedules indicated above have an impact on EMT.
Author Response
Thank you for the time spend on our manuscript. We have made the following changes.
We added the Supplement Figure 7 and 8 showing exemplary images of the colony formation assay as well as flow cytometry histograms as suggested. We included the following sentences under material and methods:
Line 491/2: Exemplary images of CFA are shown in Suppl. Figure. 7.
Line 506/7: Exemplary histograms of flow cytometry are shown in Suppl. Figure. 8.
2) The expression of EMT markers (E/N-cadherin, vimentin, etc.) is highly desirable to reveal whether the treatment schedules indicated above have an impact on EMT.
The EndoMT process is indeed very interesting and the current focus of our research. The experiments are not yet completed. However, we can provide the following insights into our research as reviewers. We have found evidence that the upregulation of EndoMT proteins, here for ACTA2/ α-smooth muscle actin and MMP2, varies between radiation qualities and in a dose-dependent manner. This effect also appears to occur with combined treatment, but with varying intensity depending on the Dox regimen. We are currently in the process of confirming our results in more complex models (3D, in ovo). Thank you again for your feedback.